# Characterization of *Steinernema feltiae* (Rhabditida: Steinernematidae) Isolates in Terms of Efficacy against Cereal Ground Beetle *Zabrus tenebrioides* (Coleoptera: Carabidae): Morphometry and Principal Component Analysis

**DOI:** 10.3390/insects14020150

**Published:** 2023-02-01

**Authors:** Joanna Matuska-Łyżwa, Barbara Wodecka, Wiesław Kaca

**Affiliations:** 1Department of Microbiology, Institute of Biology, Faculty of Natural Sciences, Jan Kochanowski University, 7 Uniwersytecka St, 25-406 Kielce, Poland; 2Faculty of Law and Social Sciences, Jan Kochanowski University, 15 Uniwersytecka St, 25-406 Kielce, Poland

**Keywords:** entomopathogenic nematodes, *Steinernema feltiae*, *Zabrus tenebrioides*, principal component analysis, cereal ground beetle, local adaptation

## Abstract

**Simple Summary:**

*Zabrus tenebrioides* (cereal ground beetle) is one of the main pests of cereals worldwide and is predicted to cause serious damage to Polish crops in the near future. A potentially effective method of biological control of this pest is the parasitism of beetle larvae by entomopathogenic nematodes. This study assessed the effectiveness of local isolates of *Steinernema feltiae* against *Z. tenebrioides* larvae under Polish field conditions, with at least 90% persistence of infectivity after 60 days in the soil. The differences in biological activity among the isolates toward the host were evaluated in terms of morphometry through principal component analysis.

**Abstract:**

One of the most dangerous pests of cereals is *Zabrus tenebrioides* and, in Poland, it is becoming a serious pest. Entomopathogenic nematodes (EPNs) seem to be a very promising, biological control agent for this pest. Native EPN populations are well adapted to local environmental conditions. The current study characterized three Polish isolates of the EPN *Steinernema feltiae*, which differed in their effectiveness against *Z. tenebrioides*. In the field, isolate iso1Lon reduced the pest population by 37%, compared with 30% by isolate iso1Dan and 0% by the iso1Obl isolate; the number of plants damaged by *Z. tenebrioides* in the presence of the different isolates reflected the results in terms of the decrease in pest population size. After incubation in the soil for 60 days, recovered EPN juveniles of all three isolates were able to infect 93–100% of the test insects, with isolate iso1Obl again showing the lowest effectiveness. The juveniles of isolate iso1Obl were also morphometrically distinct from the other two isolates, as revealed by principal component analysis (PCA), which helped to distinguish the EPN isolates. These findings showed the value of using locally adapted isolates of EPNs; two of the three isolates randomly selected from Polish soil outperformed a commercial population of *S. feltiae*.

## 1. Introduction

Larvae of the genera *Steinernema* and *Heterorhabditis* are insect parasites (entomopathogenic nematodes (EPNs)) and are used commercially as effective biocontrol agents against a number of plant pests [1,2]. Pests susceptible to EPN parasitism occur in a wide range of crops, ranging from greenhouse to field crops [3,4].

Cereals are the main crop sources of agricultural food all over the world, with more than 50% of the daily human energy consumption coming from cereals [5]. Compared with other European Union countries, Poland is one of the leading grain producers [6]. Many environmental and agronomic factors influence the production of cereals, including temperature, humidity, crop rotation, soil characteristics, the presence of crop pests and pathogens, and even the COVID-19 pandemic [7,8,9,10].

Achieving a high yield of high-quality grain requires many inputs. The presence of pests not only causes direct yield losses; however, by causing damage to plant tissues, they facilitate the penetration of phytopathogenic viruses, bacteria, or fungi, causing diseases, which, in turn, further reduce the quantity and quality of the grain obtained [11]. In order to reduce pest population size and protect plants, integrated pest management (IPM) is of great importance [12]. According to the International Organization of Biological and Integrated Control of Harmful Plants and Animals (IOBC), integrated crop protection involves “combating agripests using all available methods in accordance with economic, ecological, and toxicological requirements, which give priority to natural limiting factors and economic risk programs” [13].

Among the pests that cause losses in cereal crops in Poland, the most dangerous include *Sitobion avenae* (Hemiptera), *Oulema melanopus*, *Oulema gallaeciana* (both Coleoptera), *Haplothrips aculeatus*, *Limothrips cerealium* and *Limothrips denticornis* (all Thysanoptera), and *Chlorops pumilionis*, *Contarinia tritici*, *Sitodiplosis mosellana*, and *Haplodiplosis marginata* (all Diptera) [14,15,16,17,18,19,20]. Over the past decade, an increase in the damage caused by some cereal pests has been observed, such as *Oscinella frit* (fruit fly), *Delia coarctata* (wheat bulb fly), and *Zabrus tenebrioides* (cereal ground beetle), which previously were of no economic significance in Poland [21].

*Z. tenebrioides* belongs to the Carabidae family, and it is one of the most dangerous pests of agricultural crops [22]. This insect causes crop damage not only in Poland, but also in other countries [23,24,25,26]. According to plant protection data, Poland will suffer increasing damage from this pest in the near future [27]. The threshold of crop damage by *Z. tenebrioides* is one to two larvae or four damaged plants per m^2^ (in autumn) or three to five larvae or eight to ten damaged plants per m^2^ (in spring) [28].

After dark, the larvae of this beetle feed on young stalks of cereals (wheat, rye, and barley) and eat the parenchyma. They often gnaw at the base of the leaves, producing a jagged or skeletal appearance, or drag the leaf blades into tunnels in the soil. The greatest larval damage occurs in spring. Plants damaged in such a way die or grow vegetatively without forming flowering spike stalks. Adult beetles also damage the plant by feeding on developing grains. The eggs are laid in the soil from July to September; after the eggs hatch, the larvae hibernate. In spring, the larvae start feeding again, and, at the end of May, they transform into pupae, from which adults develop after about 1 month [24,25].

The control of *Z. tenebrioides* is difficult due to the nocturnal activity of this pest. The available chemical pesticides are expensive, harmful to other, nontarget organisms, and often ineffective against the target pest because, in its mature form, it may survive on other plants [23,29].

The presence of at least one stage of the pest lifecycle in the soil provides an opportunity to use a natural crop protection product, namely EPNs. Among the basic conditions for the use of such products is their ability to be infective toward beetle larvae and to survive in the soil environment occupied by *Z. tenebrioides.* Research into the taxonomy and commercialization of EPNs has been carried out in almost all parts of the world to isolate locally adapted EPN species or isolates for use in pest control [30,31]. Methods for the identification and classification of EPN species are based on morphometric and molecular data, all of which are expensive and time-consuming to collect and analyze [32].

These methods are also connected with mathematical approaches that achieve a rapid indication of the distinction between individual taxa. One of these methods is principal component analysis (PCA), which enables the identification of initial variables that affect the appearance of individual principal components that make up a homogeneous group. This method has already been used in the study of the variability of various animal species, including fish, cattle, goats, and crabs, as well as entomopathogenic nematodes [33,34,35,36,37,38]. The differences in the morphology of organisms may be related to their physiological properties and, in the case of EPNs, their insect-parasitic characteristics. It has been shown that the level of pest control may depend on the match between the optimal nematode species or isolate and the particular pest, as well as the environmental adaptation of the EPN to the local environmental conditions [39,40].

The goal of the current research was to assess the field efficacy of locally collected Polish isolates of *S. feltiae* against *Z. tenebrioides* larvae and to determine nematode invasiveness 14 and 60 days after application to soil. The relationship between the variables of biological activity toward the cereal ground beetle and the morphometric diversity of the EPN isolates was also examined, and we evaluated the value of morphometric features to achieve taxonomic differentiation.

## 2. Materials and Methods

### 2.1. Multiplication and Characterization of S. feltiae Isolates

Three *S. feltiae* isolates (iso1Obl, iso1Dan, and iso1Lon) were obtained from crop fields in Poland, and one commercial isolate (Owiplant, Owińska, Poland) was used as the control sample. The Polish isolates of *S. feltiae* were collected in the summer of 2019 from arable soils where wheat was grown [41]. The soils were representative of arable soils in Poland. The isolates came from the following regions: Częstochowa (iso1Obl), Włoszczowski (iso1Dan), and Sandomierski (iso1Lon). The tested *S. feltiae* isolates were molecularly and morphometrically identified earlier [41]. Reproduction of the test *S. feltiae* isolates was carried out on larvae of *Galleria mellonella*, greater wax moth (Lepidoptera: Pyralidae). *G. mellonella* was cultured as host material for EPNs at 20 °C on beeswax patches in ventilated polypropylene containers. The fourth larval stage of *G. mellonella*, with an average body weight of 140 mg, was used, with a dose of 50 infective juveniles (IJs)/insect larva on Petri dishes with filter paper, which were then stored in the incubation cabinet at 20 °C over a 5 day period. The cadavers of insects parasitized by nematodes were transferred into migration Petri dishes [42] (Anumbra, Šumperk, Czech Republic). The emerging juvenile nematodes were collected into tissue culture bottles, area 75 cm^2^ (Nunc EasYFlasks, Roskilde, Denmark), and stored at 4 °C. Attempts were made to distinguish the four isolates using morphometric features of nematodes at two developmental stages: infective juveniles and first-stage adult males [32]. The parameters monitored were total body length (L), maximum body width (W), distance from anterior end to excretory pore (EP), distance from anterior end to nerve ring (NR), distance from anterior end to end of pharynx (ES), tail length (T), anal body width (ABW), spicule length (SL), gubernaculum length (GL), and the ratios a = L/W, b = L/ES, c = L/T, D = (EP/ES) × 100, and E = (EP/T) × 100.

### 2.2. Experimental Site

The study of the survival of invasive EPN juveniles inoculated into soil and their effect on the biocontrol of *Z. tenebrioides* was carried out in the Świętokrzyskie voivodeship, on two adjacent areas of cultivated soil, each 0.5 ha in area (50°56′30.9″ N, 21°04′30.8″ E). The three sites from which the isolates were obtained were located approximately 100 km from the experimental site. In both cultivated areas, spring barley was the growing crop, with wheat being the crop grown the previous year. The agrochemical regime used was typical of local farms. The soil in the two sites was black earth type soil in class IIIb. The experiment was repeated twice times, with each plot representing a replicate.

### 2.3. Assessment of Cereal Ground Beetle Abundance in Experimental Site

The abundance of the pest was quantified on the basis of the presence of plant damage and the presence of beetle larvae in soil samples. On each plot, 10 sites, each with an area of 100 cm × 100 cm, were marked out, in which the number of plants damaged by *Z. tenebrioides* larvae per m^2^ was counted, and the average number of larvae per m^2^ was determined. Soil samples were taken from the sites marked with wooden frames using an Egner’s soil sampler with a diameter of 20 mm, to a depth of 30–35 cm; the uppermost 2 cm of the soil sample was discarded from each sample. The soil samples were evenly distributed over the entire surface of each plot, delimited by the frame. *Z. tenebrioides* larvae were isolated from the collected soil samples by sieving (6 mm mesh size of the sieves), and the average number of larvae per m^2^ was determined. The assessment of the number of pests and damaged barley plants per m^2^ was also carried out in plots where nematodes were applied (10^6^/m^2^, in accordance with the recommendations for the use of commercial EPN biopreparations on crops), 14 days after their application to the soil environment.

### 2.4. Detection of Local, Wild EPN in Experimental Site

The soil samples taken as described in Section 2.3 were also tested for the abundance of local, wild populations of EPNs by the trap insects method [43]. Under laboratory conditions, each replicate soil sample was thoroughly mixed to achieve homogeneity. The soil was divided between six sterile 250 mL vessels containing two *G. mellonella* larvae each. The soil samples were incubated in a thermostatically controlled cabinet (POL-EKO Aparatura, Wodzisław Śląski, Poland) at 20 °C and checked every 48 h over a 16 day period. Dead *G. mellonella* larvae were taken from the samples and placed in Petri dishes (90 mm in diameter; Anumbra, Šumperk, Czech Republic) on migration sponges in the incubator cabinet [42]. These dishes were checked daily for the emigration of EPN juveniles. After nematode juveniles were observed on the migration plates, their collection was started and continued for 10 days. The emerging juvenile nematodes were collected into tissue culture bottles, with a surface area of 75 cm^2^ (Nunc EasYFlasks, Roskilde, Denmark) and stored at 4 °C for 14 days until being used. The collected nematodes were identified by molecular analyses [32] to confirm the identity of the species.

### 2.5. EPN Inoculation in Field Studies

Field studies were carried out between the beginning of June and the end of August 2022. The nematode juveniles, multiplied and collected as described above, were applied to the field soil. The places of nematode application were localized with the use of wooden frames 100 cm × 100 cm. For each of the four nematode isolates (three test isolates plus the commercial isolate), ten inoculation frames were randomly distributed within the test plot, with frames separated by a distance of at least 6 m. The viability of the *S. feltiae* infective juveniles (IJs), suspended in tap water, was 95–100%. The IJ density and suspension volume applied to each replicate site were in accordance with the recommendations of the manufacturers of EPN plant protection products; the dose of nematodes applied to each inoculation frame was 10^6^/m^2^. Prior to nematode application, the soil was hydrated to improve soil conditions for the nematodes. The application was carried out in the evening hours on a windless day, using a backpack sprayer with a 0.5 mm nozzle diameter.

### 2.6. Study of EPN Persistence, Nematode Infectivity, and Control of Z. tenebrioides

The survival of the EPN juveniles applied to the soil was determined 14 and 60 days after application, using the trap insect method. *G. mellonella* larvae were used to determine the infectivity (and, hence, the persistence) of the test populations of nematodes of the four isolates extracted from the soil samples collected from the sites of previous nematode application. The extent of nematode infestation was determined as the percentage of *G. mellonella* larvae which were infested with nematodes. The identity of the nematode juveniles was confirmed by the molecular method [32,41] to confirm the identity of the isolate previously introduced into the soil. A total of 96 larvae (100%) were used for each isolate in each plot to calculate the frequency of nematode infestation. After collecting soil samples from the nematode application sites to quantify the persistence of the EPN populations, the abundance of *Z. tenebrioides* and the number of plants damaged by this pest were determined 14 days after application of the nematodes as described earlier.

### 2.7. Statistical and Data Analysis

For statistical analyses of the biological parameters, the data from each of the 10 inoculation frames for the same isolate per plot were treated as one sample, with the two plots acting as replicates. The Shapiro–Wilk test was performed to determine whether the data approximated to a normal distribution. The pest frequency and plant damage data prior to nematode application were found to be non-normally distributed. Data after application of nematodes to the soil was non-normally distributed; hence, the Kruskal–Wallis test was used to analyze any differences among the efficacies of the nematode isolates.

Principal component analysis (PCA) was carried out to analyze the morphometric diversity [44,45]. In order to check whether PCA would be appropriate, the variable correlation matrices and the Kaiser–Mayer–Olkin coefficient (KMO) were determined, and the Bartlett test was performed.

All calculations, tests and graphs were made in the R program [46] using the following packages: “dplyr” [47], “ggbiplot” [48], “factoextra” [49], and “psych” [50]. The significance level, α=0.05, was adopted as the threshold for significance.

## 3. Results

### 3.1. Presence of Local EPN Populations in the Experimental Site

Studies showed that no local, wild populations of EPNs were present in the two experimental plots.

### 3.2. Assessment of the Frequency of Z. tenebrioides in the Experimental Site

In the experimental site, the average number of *Z. tenebrioides* larvae was 2.925 larvae per m^2^, close to the lower limit of the damage threshold. With respect to the number of damaged plants, high pest activity was not demonstrated, with the average number of plants damaged by *Z. tenebrioides* being 4.725 per m^2^ (Table 1).

### 3.3. Assessment of the Efficacy of the Tested EPN Isolates against Z. tenebrioides

After application of the nematodes of the four EPN isolates to the soil, it was observed that, 14 days after their application, in the case of three of the nematode isolates, the mean number of both live Z. *tenebrioides* larvae and damaged plants decreased significantly (Table 1 and Table 2).

The iso1Lon isolate proved to be the most effective at pest elimination, as the average number of the live pest larvae was reduced by 37%. Although the number of plants damaged by *Z. tenebrioides* before treatment was not high, it decreased by 42% after application of the iso1Lon isolate. Equally satisfactory insecticidal effectiveness against *Z*. *tenebrioides* was observed in the case of comparing the iso1Dan isolate, which reduced the pest population by 30% and the number of damaged plants by 40%. The Iso1Obl isolate proved to be ineffective in controlling *Z. tenebrioides* as it reduced neither the number of pests nor the number of plants damaged by them (Table 1 and Table 2).

The Shapiro–Wilk test showed that all *p*-values were less than 0.05; therefore, the distributions of the analyzed data were not normal (Table 3). The Mann–Whitney U test was used to determine which samples differed from each other (Table 4). On this basis, it was found that the iso1Obl isolate was statistically different from the others for both the number of *Z. tenebrioides* larvae and the number of plants damaged by this pest.

### 3.4. EPN Persistence and Infectivity in the Soil

When assessing the persistence of the studied nematode isolates in the soil environment with *Z. tenebrioides*, it was shown that, 14 days after nematode application, two isolates (iso1Dan, iso1Lon) were 100% persistent (Table 5), with all the trap insects (*G. mellonella*) dying after contact with soil taken from the application sites of these nematode isolates.

The lowest persistence (number of insects killed) was recorded in soil samples containing nematodes from the iso1Obl isolate 14 or 60 days after application to the soil. The iso1Lon isolate showed the greatest persistence and effectiveness against test insects; 60 days after application, the nematodes killed all *G. mellonella* larvae with which they were incubated in vitro. The iso1Dan isolate showed similar persistence and effectiveness (Table 5).

### 3.5. Distinction among EPN Isolates Using Morphometric Features and PCA

Analyzing the determined variable correlation coefficient matrices (Table 6 and Table 7), the Kaiser–Mayer–Olkin coefficient (KMO), and the p-value in the Bartlett test (Table 8), we concluded that PCA was appropriate and brought about the intended results.

For both infective juveniles and first-stage males, the value from the Bartlett test allowed acceptance of the hypothesis that there was a significant difference between the correlation matrix and the identity matrix obtained, i.e., a significant correlation between the variables. The KMO obtained coefficient in both cases was high, greater than 0.5, allowing for the execution of PCA for the morphometric features analyzed (Table 8).

The eigenvalues obtained from PCA indicate that, in the case of infective juveniles, the first two principal components described the data well. The eigenvalue for the first component was 4.99 and the percentage of the variance it explained was 41.55%. The second component explained much less of the variance, i.e., 16.82%, and its eigenvalue was 2.02. For adult males, the two principal components also described the data well. The eigenvalue for the first component was 3.28 and the percentage of the variance it explained was 25.24%, whereas the second component explained less variance (20.93%), and its eigenvalue was 2.72. When applying the Kaiser criterion in both cases, the interpretation should take into account the first five components, because, for each of them, the eigenvalues were greater than 1 (Table 9).

When analyzing the scree plots for infective juveniles, it can be observed that the slope line turned into a horizontal line from the seventh main component onward (Figure 1). According to this indication, it can be concluded that combining the first eight components explained about 99.97% of the variance (Table 9). The plot of the adult male scree plot led to a similar conclusion, except that the first eight components together explained about 99.94% of the variance (Figure 2, Table 9).

When assessing the principal components, it was observed that, cumulatively, the first two components explained 58.37% (for juveniles) and 46.16% (for males) of the total variance (Figure 3 and Figure 4). The intensity of color of the arrows reflects the influence of the variables on the principal components. The vectors representing the original variables did not extend to the edges of the unit circle; thus, they were all moderately represented by the first two principal components making up the coordinate system. When analyzing the angles between the juvenile vectors, it can be noted that the variables total body length and c (r=0.938), total body length and b (r=0.924), and b and c (r=0.905) were strongly positively correlated, whereas the variables distance from anterior end to end of pharynx and b (r=0.0314), maximum body width and D (r=0.0673), tail length and D (r=0.0837), and anal body width and a (r=0.0900) were not significantly correlated. On the other hand, for males, there was a strong significant correlation between the variables total body length and b (r=0.889), maximum body width and a (r=−0.788), total body length and c (r=0.783), b and c (r=0.766), and tail length and E (r=−0.733), whereas spicule length and c (r=−0.00458), maximum body width and c (r=0.00500), distance from anterior end to excretory pore and a (r=−0.00531), maximum body width and b (r=0.00616), and gubernaculum length and b (r=−0.00974) were not significantly correlated (Table 6 and Table 7).

By analyzing the grouping of the population, one can observe how individual observations formed groups, depending on the isolate. For the infective juveniles, the group for the iso1Obl isolate was clearly distinguished; for the adult males, the iso1Dan isolate formed a distinctly separate group. Unfortunately, the other groups overlapped. Additionally, the plotted vectors representing the primary variables (similar to Figure 3 and Figure 4) indicate that the strongest correlation was between the variables total body length and c for infective juveniles and between the variables total body length and b for first-stage adult males. The weakest correlation was between distance from the anterior end to the end of the pharynx and b for infectious juveniles and between spicule length and c for first-stage adult males (Figure 5 and Figure 6).

## 4. Discussion

The results of field studies of control measures against pests often differ from the results of the corresponding laboratory-based studies [51]. Therefore, to improve the effectiveness of the biocontrol organisms used, multidisciplinary research is necessary, leading to a reduction in costs and an increase in the efficacy of the final product. This paper presents mathematical analysis of the biological differences among various *S. feltiae* isolates, followed by results from where this isolate was obtained, from studies on the infectivity and persistence of the different isolates against *Z. tenebrioides* in the natural environment (nonsterile soil).

The effectiveness of EPNs in the soil depends on many factors. One of them is the selection of an appropriate nematode for control of a specific pest species under prevailing environmental conditions [52,53]. In the current study, the soil type, type of crop, and the previous crop were determined, as well as the number of *Z. tenebrioides* larvae per m^2^, and the soil was examined for the presence of local, native populations of EPNs. Reports of using local EPN isolates to control *Zabrus* spp. in Turkey [54] opened up the possibility of attempting a similar strategy in Poland. No local populations of EPN were found in the experimental site, but the specific economic damage threshold of *Z. tenebrioides* in Poland indicated an urgent need for research into the effective control of this pest.

Analysis of the effectiveness of the isolates of *S. feltiae* against *Z. tenebrioides* showed differences between the isolates. The iso1Lon isolate reduced the pest population by 35%, whereas the iso1Obl isolate turned out to be completely ineffective against this insect pest species. Similar inter-isolate relationships were observed for pest control parameters number of live *Z. tenebrioides* larvae and the number of plants damaged by *Z. tenebrioides*. In the plots where the iso1Lon isolate was applied, there was a decrease in the number of damaged plants by 41% on average.

In the case of the iso1Obl isolate, it was observed that, where it was applied, the average number of damaged plants even increased, relative to the control site, confirming the lack of entomopathogenic effectiveness of this isolate. The remaining two isolates were characterized by a slightly lower insecticidal effectiveness than in the case of the iso1Lon isolate. These in-soil results prove that two of the three randomly selected isolates proved to be more effective than the commercial isolate when applied under environmental conditions similar to those from which the wild EPN isolates were isolated. Other studies have provided evidence that EPNs show considerable variation in terms of biological activity, host selection, and tolerance to various environmental conditions [55]. This finding on the effectiveness of local EPN isolates against *Zabrus* spp. supports research described earlier [54]. The subject of the biology of local EPN populations has been discussed for a long time. Recent studies have shown that local populations of EPNs are highly adapted to the environmental conditions from which they were isolated [55]. Nematode isolates studied in the current research were isolated and used in the same country and in similar environments, which could have influenced the results obtained.

The adaptive abilities of the studied nematodes could also have an impact on the persistence of the population in the target environment. The three tested nematode isolates recovered 14 or 60 days after application to the soil were able to infect 93–100% of the test insects. The iso1Obl isolate also showed the lowest persistence in this study, killing only 83–92% of the hosts. This shows that the isolates retained their infectivity over a period of at least 2 months, although earlier research reports have indicated that EPNs introduced into the soil could persist for between 1 and 3 years [54,56].

Research on the morphometric diversity of entomopathogenic nematodes provided valuable information on their geographic and ecological requirements. The morphometry of these organisms may be influenced by various factors, such as geographical origin, habitat, or host species [57,58]. This study described the differentiation of *S. feltiae* isolates from different locations, but within the same geographical region. PCA for the morphometric features of infective juveniles showed that the isolate iso1Obl differed the most from the other isolates, reflecting the relatively low bioactivity of this isolate. Mathematical analyses of the biological activity of this isolate against *Z. tenebrioides* and *G. mellonella* also showed its distinctiveness. The biological and morphometric diversity of the iso1Obl isolate larvae was confirmed by another study [41], which reported a different level of biological activity of this isolate against *G. mellonella*.

EPNs are morphometrically very diverse organisms. PCA confirmed that the morphometric features of nematodes used for identification indicated in the literature [32] are very important features in the classification of these organisms. In earlier work, PCA was used for the classification of Argentinian *Heterorhabditis* species, where the method correctly classified different species into different groups [59]. In turn, PCA was used by another research group to show that the method effectively differentiated *Heterorhabditis baujardi* from *Heterorhabditis indica* [60]. Similar conclusions were reached by a team using PCA to differentiate morphometric traits in adult males and juveniles to identify differences in *Steinernema hermaphroditum* populations, finding some variability [61]. Furthermore, in the current study, the isolates were correctly classified into different groups, whereas the authors of [37] found that a combination of molecular technique and classical morphological studies was a useful tool for assessing the biodiversity of Steinernematidae nematodes and could also be useful for determining differences in pathogenicity toward insect pests [37].

## 5. Conclusions

The research conducted proved that two out of three randomly selected local EPN isolates from Poland turned out to be more effective against *Z. tenebrioides* than a commercial population under Polish conditions. This result confirms the relative value of using local EPN isolates. These isolates differed in their infectivity against *Z. tenebrioides* and in their persistence to infect the host after re-isolation following a period in the soil. PCA confirmed the biological and morphometric differences between the isolates and confirmed the significance of the features important for the identification and taxonomic classification of *S. feltiae*.

## Figures and Tables

**Figure 1 insects-14-00150-f001:**
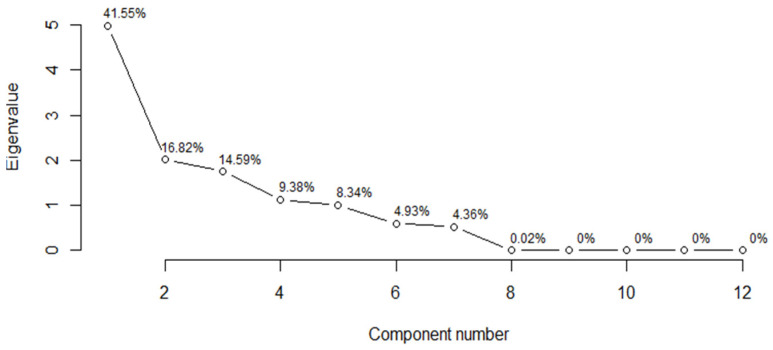
Scree plots of the ordered eigenvalues based on infective juveniles.

**Figure 2 insects-14-00150-f002:**
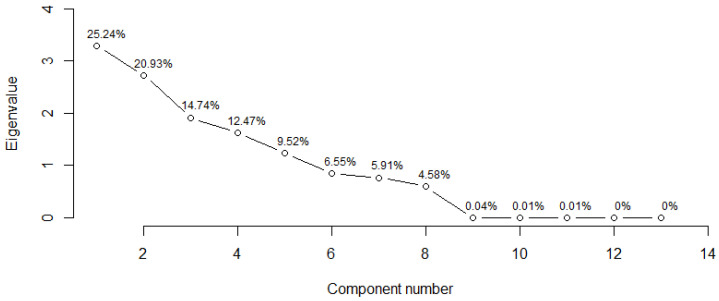
Scree plots of the ordered eigenvalues based on first-stage adult males.

**Figure 3 insects-14-00150-f003:**
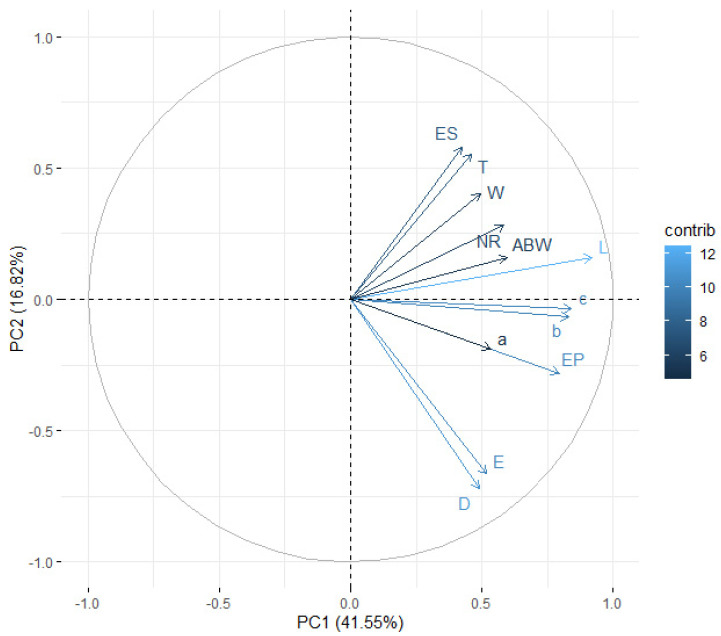
Principal component analysis of morphometric variables for infective juveniles.

**Figure 4 insects-14-00150-f004:**
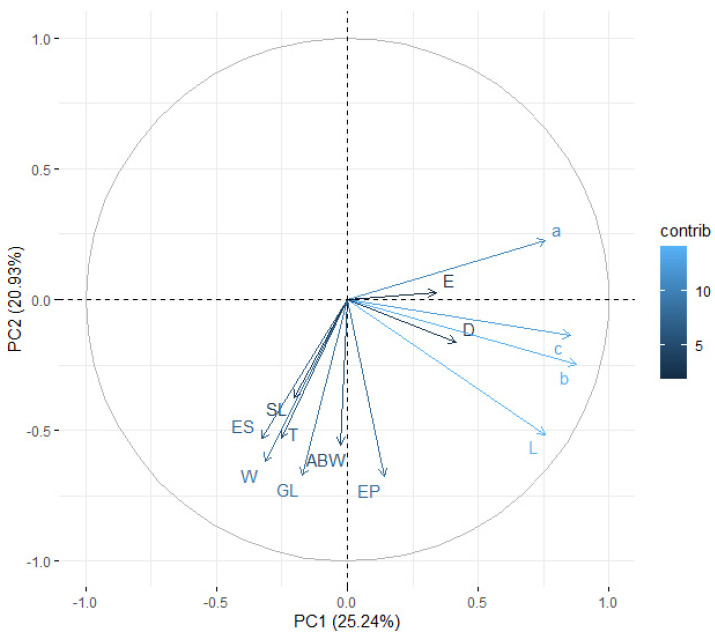
Principal component analysis of morphometric variables for first-stage adult males.

**Figure 5 insects-14-00150-f005:**
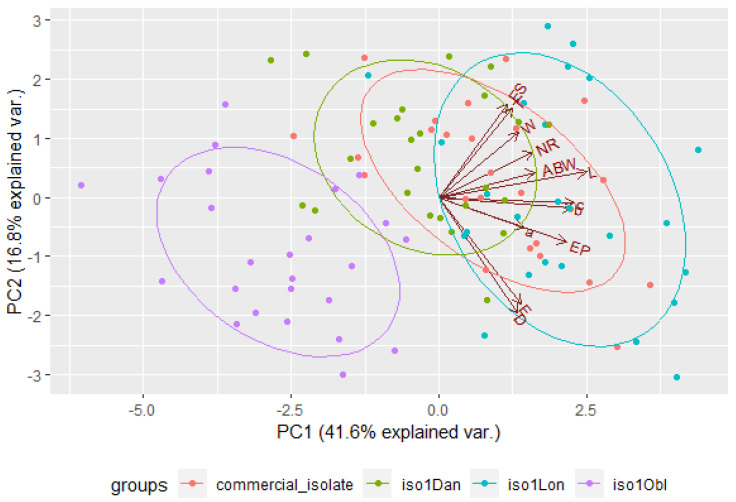
Principal component analysis for infective juveniles—individuals.

**Figure 6 insects-14-00150-f006:**
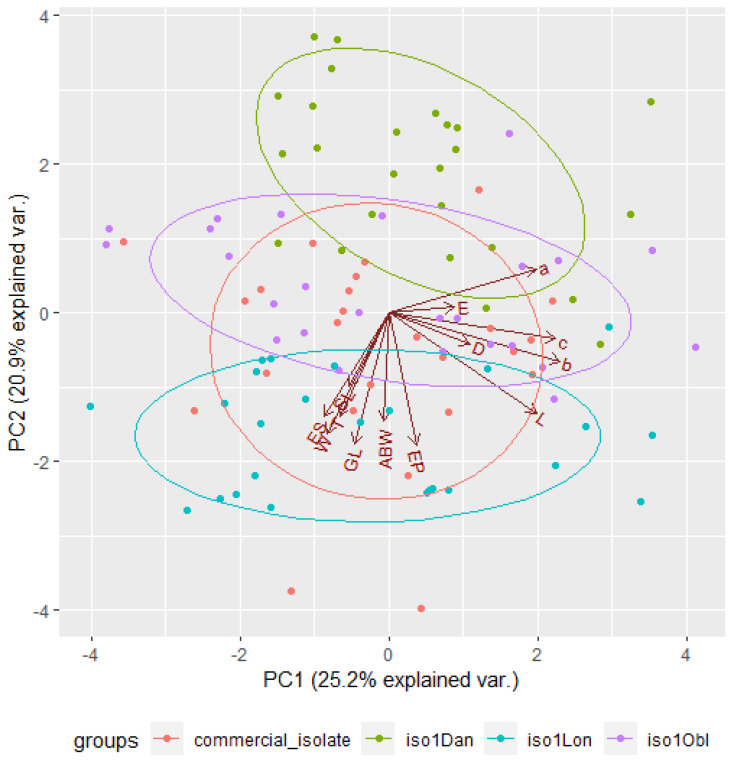
Principal component analysis for first-stage adult males—individuals.

**Table 1 insects-14-00150-t001:** Student’s *t*-test for numbers of live *Z. tenebrioides* larvae and damaged plants per m^2^ before the application of nematodes.

Parameter	*p*-Value Shapiro–Wilk Test	Median	Mean	Standard Deviation (SD)
Live larvae of *Z. tenebrioides*	0.08991	3	2.925	0.6857
Plants damaged by *Z. tenebrioides*	0.09016	5	4.725	1.2401

**Table 2 insects-14-00150-t002:** Mean number of live *Z. tenebrioides* larvae and damaged plants per m^2^ after 14 day exposure to four isolates of *S. feltiae* nematodes.

Nematode Isolates	Parameter	Median	Mean	SD
Commercial isolate	Live larvae	2	2.25	0.9105
	Plants damaged	3	2.95	0.9987
iso1Dan	Live larvae	2	2.05	0.6863
	Plants damaged	3	2.85	0.7452
iso1Lon	Live larvae	2	1.85	0.7452
	Plants damaged	3	2.75	0.7864
iso1Obl	Live larvae	3	3.25	0.9666
	Plants damaged	5	5.25	1.0196

**Table 3 insects-14-00150-t003:** *p*-Values in the Shapiro–Wilk test.

Nematode Isolates	No. Live Larvae of *Z. tenebrioides*	No. Plants Damaged by *Z. tenebrioides*
Commercial isolate	0.0158	0.0022
iso1Dan	0.0011	0.0012
iso1Lon	0.0012	0.0094
iso1Obl	0.0193	0.0110

**Table 4 insects-14-00150-t004:** *p*-Values in the Mann–Whitney U test for the number of live larvae (values above the main diagonal of the table) and the number of damaged plants (values below the main diagonal of the table).

	Commercial Isolate	iso1Dan	iso1Lon	iso1Obl
Commercial isolate	—	0.5382	0.1701	0.0031
iso1Dan	0.9425	—	0.3690	0.0002
iso1Lon	0.7398	0.7697	—	0.0001
iso1Obl	0.0000	0.0000	0.0000	—

**Table 5 insects-14-00150-t005:** Percentage of *G. mellonella* larvae killed by nematodes of different EPN isolates after 14 or 60 days in the soil.

Days	Commercial Isolate	iso1Dan	iso1Lon	iso1Obl
14 days	97%	100%	100%	92%
60 days	93%	99%	100%	83%

**Table 6 insects-14-00150-t006:** Correlation coefficients among morphometric traits of *S. feltiae* infective juveniles for all isolates. All coefficients that are statistically significant (p<0.05) are marked in red.

	L	W	EP	NR	ES	T	ABW	a	b	c	D	E
L	1	—	—	—	—	—	—	—	—	—	—	—
W	0.484	1	—	—	—	—	—	—	—	—	—	—
EP	0.531	0.306	1	—	—	—	—	—	—	—	—	—
NR	0.418	0.336	0.470	1	—	—	—	—	—	—	—	—
ES	0.409	0.324	0.392	0.401	1	—	—	—	—	—	—	—
T	0.441	0.305	0.382	0.427	0.409	1	—	—	—	—	—	—
ABW	0.422	0.398	0.474	0.446	0.272	0.363	1	—	—	—	—	—
a	0.627	−0.375	0.289	0.141	0.149	0.190	0.090	1	—	—	—	—
b	0.924	0.396	0.416	0.291	0.031	0.307	0.346	0.625	1	—	—	—
c	0.938	0.418	0.440	0.301	0.298	0.102	0.323	0.622	0.905	1	—	—
D	0.230	0.067	0.726	0.179	−0.347	0.084	0.277	0.180	0.395	0.221	1	—
E	0.255	0.115	0.782	0.208	0.132	−0.276	0.250	0.170	0.225	0.389	0.699	1

**Table 7 insects-14-00150-t007:** Correlation coefficients among morphometric traits of first-stage male *S. feltiae* for all isolates. All coefficients that are statistically significant (p<0.05) are marked in red.

	L	W	EP	ES	T	ABW	SL	GL	a	b	c	D	E
L	1	—	—	—	—	—	—	—	—	—	—	—	—
W	0.138	1	—	—	—	—	—	—	—	—	—	—	—
EP	0.261	0.206	1	—	—	—	—	—	—	—	—	—	—
ES	0.140	0.266	0.385	1	—	—	—	—	—	—	—	—	—
T	0.206	0.187	0.313	0.250	1	—	—	—	—	—	—	—	—
ABW	0.183	0.266	0.255	0.168	0.216	1	—	—	—	—	—	—	—
SL	0.015	0.217	0.115	0.270	0.035	0.144	1	—	—	—	—	—	—
GL	0.149	0.312	0.351	0.335	0.313	0.289	0.288	1	—	—	—	—	—
a	0.493	−0.788	−0.005	−0.164	−0.048	−0.132	−0.184	−0.180	1	—	—	—	—
b	0.889	0.006	0.075	−0.328	0.078	0.100	−0.107	−0.010	0.549	1	—	—	—
c	0.783	0.005	0.037	−0.035	−0.445	0.031	−0.005	−0.061	0.484	0.766	1	—	—
D	0.119	−0.045	0.593	−0.515	0.069	0.089	−0.126	0.034	0.140	0.354	0.065	1	—
E	−0.015	−0.032	0.415	0.034	−0.733	−0.030	0.048	−0.052	0.040	−0.024	0.450	0.359	1

**Table 8 insects-14-00150-t008:** p-Values in Bartlett’s test and Kaiser–Meyer–Olkin (KMO) criterion.

	p-Value	KMO
Infective juveniles	<2.2 $\times \;10^{-16}$	0.6515
Adult first-stage males	<2.2 $\times \;10^{-16}$	0.5471

**Table 9 insects-14-00150-t009:** Initial eigenvalues from PCA for infective juveniles and first-stage adult males.

Principal Component	Infective Juveniles			First-Stage Adult Males		
	Total	% of Variance	Cumulative %	Total	% of Variance	Cumulative %
1	4.9861	41.5509	41.5509	3.2807	25.2360	25.2360
2	2.0182	16.8181	58.3690	2.7207	20.9288	46.1648
3	1.7501	14.5845	72.9535	1.9168	14.7448	60.9096
4	1.1261	9.3840	82.3375	1.6207	12.4673	73.3768
5	1.0005	8.3376	90.6751	1.2376	9.5198	82.8966
6	0.5920	4.9331	95.6081	0.8519	6.5534	89.4500
7	0.5231	4.3590	99.9671	0.7680	5.9079	95.3578
8	0.0027	0.0222	99.9893	0.5960	4.5849	99.9428
9	0.0005	0.0038	99.9931	0.0047	0.0362	99.9790
10	0.0004	0.0032	99.9963	0.0014	0.0109	99.9900
11	0.0003	0.0022	99.9985	0.0007	0.0055	99.9954
12	0.0002	0.0015	100	0.0004	0.0028	99.9982
13	—	—	—	0.0002	0.0018	100

## Data Availability

The datasets are available on reasonable request to the corresponding author.

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
