# Peer review of "Characterization of Steinernema feltiae (Rhabditida: Steinernematidae) Isolates in Terms of Efficacy against Cereal Ground Beetle Zabrus tenebrioides (Coleoptera: Carabidae): Morphometry and Principal Component Analysis"

_insects, 2023, doi:10.3390/insects14020150_

Round 1

Reviewer 1 Report

The Manuscript [insects-2176042] entitled [Characterization of Steinernema feltiae (Rhabditida: Steinernematidae) Isolates in Terms of Efficacy Against Corn Ground Beetle Zabrus tenebrioides (Coleoptera: Carabidae), Morphometry and Principal Component Analysis] characterized three Polish isolates of the EPN Steinernema feltiae. Results indicated that the 3 isolates were differed in their effectiveness and in their morphometric.

 Introduction and results are introduced and written well. Meanwhile, methods need more clarifications

Major comments:

1-     Line 127: write in exact all locations with longitude and latitude

2-     Line 131: where these 2 replicates?. Are data were combined?

3-     Line 136: [On each plot, 10 sites, each….] it is not clear. Are the 10 sites replicates? What is the design in the field?

4-     Line 146: [nematodes were applied (106/m2)]. From where this recommended dose?

5-     Lines 168-169: This method needs more clarification and should add reference

6-     Line 162-193: This sub-section should be more clear with the DNA isolation, PCR conditions with the reference (ref. not included in ref. list

7-     Accession numbers of these isolates should be added

8-     Table 5: where SE or SD of these data?. Where the statistic of them?

Minor comments:

1-      Line 19: use [among] instead of [between], also, in line 208

2-      Line 64: [ in recent years] and in line 92 [Currently]; these words are not suitable with the years of cited references.

3-      Line 122: add family and order of Galleria mellonella, then, abbreviate the genus in line 151

4-      Line 201: change the title to [Statistical analysis]

5-      Table 2: delete the column of [Parameter] and indicate it in the title of the table

6-      Tables 6 and 7: all comparison were repeated. Keep only the left-down side and add [-] in the right-upper side.

Author Response

Dear Reviewer.

Thank you very much for the comments. We appreciate the time and effort that you dedicated to providing feedback on our manuscript and are grateful for the insightful comments on and valuable improvements to our paper. We hope that our answers and corrections will be satisfactory.

Major comments:

  • Line 127: write in exact all locations with longitude and latitude

Your remark is most accurate. Correction is included in the manuscript.

  • Line 131: where these 2 replicates?. Are data were combined?

Thank you for your remark. Correction is included in the manuscript. The experiment was repeated twice, with each plot representing a replicate.

For statistical analyses of the biological parameters, the data from each of the ten inoculation frames for the same isolate per plot were treated as one sample, with the two plots acting as replicates.

  • Line 136: [On each plot, 10 sites, each….] it is not clear. Are the 10 sites replicates? What is the design in the field?

Your remark is most accurate. Correction is included in the manuscript. Ten frames (100 cm × 100 cm) were randomly distributed within each test plot, with frames separated by a distance of at least 6 m.

  • Line 146: [nematodes were applied (106/m2)]. From where this recommended dose?

Thank you for your remark. Correction is included in the manuscript. The applied dose of 106/m2, in accordance with the recommendations for the use of commercial EPN biopreparations on crops.

  • Lines 168-169: This method needs more clarification and should add reference

Your remark is most accurate. Correction is included in the manuscript.

  • Line 162-193: This sub-section should be more clear with the DNA isolation, PCR conditions with the reference (ref. not included in ref. list

Thank you for your remark. The reference included. This correction is included in the manuscript.

  • Accession numbers of these isolates should be added

Thank you for your remark. The primers used in the research [41] allow a short fragment of ITS to be amplified. EPN isolates were assigned to species, but not for sub-species. In future studies, we plan to perform a phylogenetic analysis downstream of the species then the isolates will be re-sequenced for a wider range and entered into the GenBank database.

  • Table 5: where SE or SD of these data?. Where the statistic of them?

Thank you for your suggestions. Table 5. has showed percentage of G. mellonella larvae killed by nematodes of different EPN isolates after 14 or 60 days in the soil. A total of 96 larvae (100%) were used for each isolate in each plot.

Minor comments:

  • Line 19: use [among] instead of [between], also, in line 208
  • Line 64: [ in recent years] and in line 92 [Currently]; these words are not suitable with the years of cited references.
  • Line 122: add family and order of Galleria mellonella, then, abbreviate the genus in line 151
  • Line 201: change the title to [Statistical analysis]
  • Table 2: delete the column of [Parameter] and indicate it in the title of the table
  • Tables 6 and 7: all comparison were repeated. Keep only the left-down side and add [-] in the right-upper side.

Thank you for your remarks. All corrections are included in the manuscript.

Best regards,

Authors

Reviewer 2 Report

The work is interesting but written in a confusing way, the objective of the work is not well defined and becomes clear only after careful reading of the whole article (even the title should be revised).Line 19: The term "characterized" must be replaced with "evaluated".Line 20: replace "and" with "through".The sentence of lines 33-35 seems to refer to morphometric differences rather than to the results of the control of Z. tenebrioides, as indeed it isLine 87: I would replace "in the form" with "which"Lines 92-96: PCA or any other statistical method cannot serve to distinguish taxa, it can help to study and signify taxa variability.Lines 107-112: cumbersome and unclear.The citation from line 123 must be moved to line 117 or must be accompanied by the explanation that the taxa have been determined as S. feltiae in the cited paper.Line 201: it would be better to replace with: Statistical analysis.Line 206: replace Wilcox test with Wilcoxon Mann-Whitney test or with Mann-Whitneu U test, if only part of the test is applied, and standardize throughout the text.Tab. 1 Fig. 1 and Fig. 2 are useless, can be eliminated and the text needs to be revised.Lines 356-366: The correlation in PCA between L and c, L and b, Pharynx length and b, MBW and a, T and E is obvious; b, c, etc.  are relationships that use the other parameter with which they are correlated in the numerator or in the denominator.Lines 365-398 unclear sentence.Many of the things written in the discussion have already been said in the introduction and in the results.Line 427: the bibliographic citations are to be eliminated, if they are to be kept it is necessary to explain their contribution.Lines 440-441: the study does not describe the difference between the isolates but analyzes the morphometric differences with PCA.Line 458: There is only one species, therefore replace "were" with "was" or "the species" with "the isolates"

Author Response

Dear Reviewer.

Thank you very much for the comments. We appreciate the time and effort that you dedicated to providing feedback on our manuscript and are grateful for the insightful comments on and valuable improvements to our paper. We hope that our answers and corrections will be satisfactory.

Line 19: The term "characterized" must be replaced with "evaluated".

Your remark is most accurate. Correction is included in the manuscript. 

Line 20: replace "and" with "through".

Your remark is most accurate. Correction is included in the manuscript. 

The sentence of lines 33-35 seems to refer to morphometric differences rather than to the results of the control of Z. tenebrioides, as indeed it is

Thank you for your suggestions.

Line 87: I would replace "in the form" with "which"

Your remark is most accurate. Sentence corrected.

Lines 92-96: PCA or any other statistical method cannot serve to distinguish taxa, it can help to study and signify taxa variability.

Your remark is most accurate. Correction is included in the manuscript. Lines 107-112: cumbersome and unclear.Thank you for your remark. This paragraph corrected. 

The citation from line 123 must be moved to line 117 or must be accompanied by the explanation that the taxa have been determined as S. feltiae in the cited paper.

Your remark is most accurate. Correction is included in the manuscript.

Line 201: it would be better to replace with: Statistical analysis.

Thank you for your remark. Correction is included in the manuscript.

Line 206: replace Wilcox test with Wilcoxon Mann-Whitney test or with Mann-Whitneu U test, if only part of the test is applied, and standardize throughout the text.

Your remark is most accurate. Correction is included in the manuscript.

Tab. 1 Fig. 1 and Fig. 2 are useless, can be eliminated and the text needs to be revised.

Thank you for your suggestions.

Lines 356-366: The correlation in PCA between L and c, L and b, Pharynx length and b, MBW and a, T and E is obvious; b, c, etc.  are relationships that use the other parameter with which they are correlated in the numerator or in the denominator.

Thank you for your remark.

Lines 365-398 unclear sentence. Many of the things written in the discussion have already been said in the introduction and in the results.

Your remark is most accurate. Correction is included in the manuscript.

Line 427: the bibliographic citations are to be eliminated, if they are to be kept it is necessary to explain their contribution.

Thank you for your remark. Correction is included in the manuscript.

Best regards,

Authors

Lines 440-441: the study does not describe the difference between the isolates but analyzes the morphometric differences with PCA.

Your remark is most accurate. Correction is included in the manuscript.

Line 458: There is only one species, therefore replace "were" with "was" or "the species" with "the isolates"

Thank you for your remark. Correction is included in the manuscript.

Round 2

Reviewer 1 Report

Authors made all my comments and clarified what was previously requested